## Overview Review

coastal recession; coastal erosion; sea-level rise; shoreline change; satellite-derived shorelines; sediment budget

**Corresponding author:**
Colin D. Woodroffe;
Email: colin@uow.edu.au

‡The article has been updated since original publication. A notice detailing the change has also been published.

# Coastline changes: A reconsideration of the prevalence of recession on sandy shorelines‡

Colin D. Woodroffe[1] , Niki Evelpidou[2], Irene Delgado-Fernandez[3], David Green[4], Dhritiraj Sengupta[5], Anna Karkani[2] and Paolo Ciavola[6,7]

[1]School of Science, University of Wollongong, Wollongong, NSW, Australia; [2]Faculty of Geology & Geoenvironment, National and Kapodistrian University of Athens, Athens, Greece; [3]Department of Earth Sciences, University of Cadiz, Puerto Real, Spain; [4]AICSM, Department of Geography and Environment, School of Geosciences, University of Aberdeen, St. Mary's, Scotland, UK; [5]Plymouth Marine Laboratory, Plymouth, UK; [6]Dipartimento di Fisica e Scienze della Terra, Università di Ferrara, Ferrara, Italy and [7]CNR-IAS, Istituto per lo studio degli Impatti Antropici e Sostenibilità in ambiente marino, Oristano, Italy

## Abstract

It is often inferred that rising sea levels will result in widespread coastal recession. Erosion appeared prevalent in a worldwide compilation of evidence derived from maps and aerial photographs undertaken in the 1980s by the Commission on the Coastal Environment. Eric Bird, chair of the commission, inferred that >70% of sandy coastlines had retreated, a generalisation that has been widely cited. We reconsider these findings in respect of subsequent advances in shoreline mapping, including greater precision possible using geographical information systems and more frequent remote sensing imagery with increased spatial, spectral and temporal resolution. Satellite-derived shorelines now enable broad global and regional generalisations about shoreline position. Beaches fluctuate over a range of timescales, meaning that trends in their position are highly dependent on techniques and temporal scales adopted for monitoring. Recent global- and regional-scale shoreline assessments indicate that many sandy shorelines have been stable, and that detectable retreat has occurred on fewer beaches than previously inferred. Accretion is apparent on some coasts, particularly where engineering interventions protect or have reclaimed land. There is considerable variability in the behaviour of monitored beaches, and it is not yet possible to decipher a response to the gradual centimetre-scale rise in sea level of recent decades. Instead, we re-emphasise the several other factors that were considered to contribute to recession by the Commission, many of which relate to a change in sediment budget. To provide insights into future coastline behaviour, a better understanding of the multiple drivers on individual beaches is needed to discriminate between erosional events and longer-term trends in shoreline position.

## Impact statement

There is a widespread perception that coasts are eroding, and further recession is anticipated due to sea-level rise associated with global warming. A compilation of global evidence of coastline changes was undertaken by the Commission on the Coastal Environment in the 1970s and early 1980s, based largely on maps and aerial photography. Results were summarised in a book entitled *Coastline Changes: A Global Review.* It was estimated that 70% of the world's beaches had been retreating. Erosion is widespread on sandy beaches; however, it requires monitoring over time to determine whether there is a trend of long-term retreat. There have been considerable geospatial methodological advances since the 1980s, enabling more accurate measurements of shoreline position. Assessments based on satellite imagery, at both the global and regional levels, indicate that a far smaller proportion of unconsolidated shorelines have been retreating (limited by the resolution at which such assessments can be made). In many places, coastal infrastructure has been protected by hard or soft engineering intervention, and since 2000, substantial land reclamation has occurred. Beaches that have been monitored for decades indicate the complexity of shoreline behaviour as they respond to changing wave conditions and the impact of large storms, masking their response to the gradual rise of sea level. Many factors, recognised in the earlier review, can contribute to a change in the sediment budget. Our overview reinforces the significance of the supply and transport of sand and gravel and reiterates that coastal erosion can rarely be attributed to a single causative factor, such as sea-level rise. We infer that (1) the commonly held belief that most sandy coasts are experiencing widespread, long-term recession is increasingly questionable and (2) the current impacts of sea-level rise on global shoreline trends are not yet clearly discernible, given the small magnitude of rise and the complexity of shoreline dynamics.

## Introduction

In 1972, a Working Group of the International Geographical Union (IGU), chaired by Eric Bird, began to consider the *Dynamics of Coastline Erosion*. In 1976, Bird produced a report entitled '*Shoreline changes in the British Isles during the past century*' (Bird and May 1976), building on studies undertaken by the Royal Commission on Coastal Erosion in Britain in the early years of the twentieth century. The IGU Working Group became the *Commission on the Coastal Environment* (1976–1984). The Commission, under Bird's chairmanship, brought together information from over 200 correspondents representing 127 countries and summarised this in a book entitled '*Coastline Changes: A Global Review*'. These wide-ranging studies concluded that 'erosion has been more extensive than deposition around the world's coastline in recent decades, especially on low-lying sandy coasts' (Bird 1985, p. ix). In a follow-up publication, Bird stated that more than 70% of the world's sandy coastlines had retreated, and <10% had prograded (Bird 1987). This generalisation at a global scale has been frequently cited, as an increasing number of researchers have examined the effects of rising sea levels.

This study reconsiders the broad global assessment of coastline changes undertaken by the *Commission on the Coastal Environment* in the 1970–1980s. It briefly discusses methodological advances since then: the concept of coastal morphodynamics, geospatial techniques for more precise analysis primarily using aerial photography and the increasing use of satellite-derived shorelines (SDSs) to detect coastline trends.

Although the 1985 assessment included gradual retreat of rocky coasts and the dynamics of muddy coasts and coastal wetlands, these are beyond the scope of this study. Instead, we focus on sandy shorelines but exclude the trajectory of change on small sand cays and shingle islands on coral reefs, which were not covered in any detail in the Coastline Changes book. Although recent studies tend to place an emphasis on sea-level rise, many other factors contributing to the erosion of beaches were identified in the IGU project and also need to be considered. Coastal recession can rarely be attributed to any single factor. The contribution from sea-level rise is generally not yet discernible. There remain multiple challenges in forecasting medium to long-term trends in coastal behaviour. We outline other contributing factors, particularly rates and pathways of sand and gravel transport, identified during those earlier studies.

## Commission on the coastal environment and the scope of the 1985 book

As Chair of the IGU *Commission on the Coastal Environment* from 1976 to 1984, Eric Bird served as convenor for worldwide studies of the dynamics of shoreline change that contributed to '*Coastline Changes: A Global Review*' (Bird 1985). The approach appears to have been primarily driven by Bird himself, and coastal researchers from around the world with whom he corresponded and with whom, in many cases, he co-authored. The observations synthesised in the book also formed the basis for a more comprehensive compilation: *The World's Coastline* (Bird and Schwartz 1985) and the *Encyclopedia of the World's Coastal Landforms* (Bird 2010).

A major outcome of the compilation of observations from around the world was recognition of the widespread prevalence of coastal erosion, showing that the assumption that 'erosion on some sectors of sandy shorelines is balanced by deposition on other sectors' is incorrect.

## Constraints on determining coastline changes

The introductory chapter of *Coastline Changes* sets out constraints on examining evidence on which to base reconstructions of changes in shoreline position. Although called coastline changes, an important distinction was made between the coastline, which is where the land meets the sea, and the shoreline, which is the water's edge that changes over short timescales, most obviously with the tide. Sources of information available for determining changes included comparison of maps and charts, each with constraints on their accuracy, and the use of aerial photographs. Bird indicated that coastline changes could be expressed in three ways: (i) linear terms, as an advance or retreat measured at right angles to the coast; (ii) in terms of area, as the extent of land gained or lost; (iii) or in volumetric terms, as the quantity of material added to, or lost from, the shoreline (Bird 1985, p. 5).

Bird was also aware of the need to specify what indicator, or 'proxy', of the shoreline was being used. Rates of change are often expressed as an annual average, although retreat may be highly irregular, and it is important to be clear about the period over which such observations have been averaged. Use of maps or charts is constrained by the purpose of each, and the level of detail to which they were mapped, which in turn restricts the precision with which rates can be determined (Bird, 1985). Maps focus on the land, whereas charts, designed for navigation at sea, are less likely to represent the boundary of the land as accurately. Such issues are comprehensively reviewed by Monmonier (2008).

## Evidence of coastline change

Most of the book comprises a systematic review of the world's coastline, including 127 countries, described in a sequence commencing on the Arctic coast of Alaska and proceeding counter-clockwise around North and South America, contrasting the steeper cliffed western coasts with the sedimentary eastern coasts. The Arctic coasts were considered relatively stable, noting that there was little historical evidence before 1950. Brief mention was made of Caribbean islands, Greenland and Iceland.

Consideration of Europe commenced with Scandinavia, included an extensive section on the British Isles, and a country-by-country account of the coasts of the Mediterranean and Black Sea, followed by a brief synopsis of the west, south and east coasts of Africa. From Iran, coverage continued around southern and southeast Asia, Japan and to the Arctic USSR, where rapid retreat had been documented by Zenkovich (1967). Text on the Philippines, Indonesia and Papua New Guinea focused on accretion at the mouths of major rivers. The 14 pages about Australia contain several illustrated examples of accretion, but local examples of recession, and New Zealand was also described in terms of sites where progradation had been observed. This was followed by succinct descriptions of New Caledonia, Fiji, Hawaii, Tahiti and other islands in the Pacific, Atlantic and Indian Oceans, culminating in a paragraph about Antarctica.

## Factors contributing to coastline changes

Chapter 3 was entitled *Categories of coastal change*. It considers the retreat of cliffs and the accretion of deltaic coasts, as well as the effects of tectonics and volcanic activity, with a brief section on coastal wetlands. However, it is the observations made on sandy shorelines and the inferences that were drawn by Bird in the

ensuing paper (Bird 1987) on the prevalence of beach erosion that is the focus of this reappraisal.

Although there is now a widely perceived view that sea-level rise will result in coastal erosion, Bird, synthesising the *Commission on the Coastal Environment* project, emphasised the many more direct causes of beach erosion and other coastline changes, and only a brief reference was made to sea-level rise. This is not surprising, as sea-level rise as an issue for the future was only beginning to become apparent in the 1980s (Titus 1986; Hoffman et al. 1983). In his synthesis, Bird commented that 'it is widely held that a world-wide rise of sea level has taken place during the past few decades, at an average of just over a millimetre a year' (Bird, 1985, p. 169), but added that it is 'doubtful whether so small a change in the level of the oceans is sufficient to account for predominance of beach erosion, although it certainly would have been a contributing factor' (Bird 1985, p. 170).

In a summary of the implications of the Commission's findings for sandy beaches, published in 1987 in *Marine Pollution Bulletin*, Bird identified 7 situations where coastlines were prograding, but 14 factors were also identified that have contributed to the initiation or acceleration of erosion on sandy coastlines (Bird 1987). Unconsolidated beach sands and gravel are supplied to the coast primarily from rivers, eroding cliffs, from the seafloor or by wind. Accordingly, progradation was recorded from locations where these sources were delivering increased volumes of sediment to the coast, or where there was longshore delivery of sand. Progradation was also observed where there had been a relative fall in sea level (usually due to isostatic uplift as in the Gulf of Bothnia), or where sand had been artificially augmented, such as through beach nourishment (Bird, 1987).

Table 1 lists the factors that lead to the retreat of sandy coastlines, as indicated by Bird (1987, 1993). Although causes included relative 'sea-level rise', listed as number 9 in the 1987 listing, this was not especially prominent or invoked in the various studies in the synthesis. However, in his 1993 book, *Submerging Coasts*, Bird augmented the 1985 observations and identified 20 causes of beach erosion, and in this instance, he did list sea-level rise as number 1 (Bird 1993, p. 53).

Most factors in Table 1 involve some aspect of the overall sediment budget of a section of coast. The significance Bird placed on sediment supply and transport pathways can be seen in several of the various causes of beach erosion: a reduction in supply of river sediment (1), reduced delivery from cliff erosion (2), reduced supply from offshore (4) or from alterations to longshore transport (5). Similarly, Bird pointed out that sand might be lost from the system because of stabilisation of foredunes by vegetation (3), or its loss inland through aeolian processes, and subsequent cover by vegetation (7). Sand volume could also be lost through attrition or other weathering or reduction processes (11).

A second prominent set of causes of beach erosion relates to human activities: removal of sand by quarrying (6), beach adjustment following nearshore dredging (8) or response to engineering structures such as breakwaters (13). Beach retreat due to increased wave exposure (10 and 14) and a rise in water table (12) could be regarded as due to a change in climate, perhaps in response to human-induced global warming.

In describing the range of factors (summarised in Table 1) that could contribute to a trend of persistent erosion, Bird clearly acknowledged that 'no one hypothesis can account for the prevalence of beach erosion in the variety of environments around the world's coastlines' (Bird 1985, p. 174). The relative significance of

**Table 1.** Factors that have contributed to the initiation or acceleration of erosion on sandy coastlines (based on Bird 1987, 1993)

| | |
|---|---|
| 1 | Diminution of fluvial sand and shingle supply to the coast as a result of reduced runoff or sediment yield from a river catchment |
| 2 | Reduction in sand and shingle supply from eroding cliffs or shore outcrops |
| 3 | Reduction of sand supply to the shore, where dunes that had been moving from inland are stabilised |
| 4 | Diminution of sand and shingle supply washed in by waves and currents from the adjacent sea floor |
| 5 | Reduction in sand and shingle supply from alongshore sources as a result of interception |
| 6 | Removal of sand and shingle from the beach by quarrying or the extraction of mineral deposits |
| 7 | Landward drifting of dunes, notably where backshore dunes have lost their retaining vegetation cover |
| 8 | Increased wave energy reaching the shore because of the deepening of the nearshore water |
| 9 | Submergence and increased wave attack as a result of a rise in sea level |
| 10 | Increased wave attack due to climate change that has produced a higher frequency, duration or severity of storms |
| 11 | Diminution in the volume and/or calibre of the beach and nearshore material as a result of attrition or weathering |
| 12 | A rise in the water table within the beach due to increased rainfall or local drainage modification |
| 13 | Increased losses of sand and shingle alongshore as a result of a change in the angle of incidence of waves |
| 14 | Intensification of wave attack as a result of the lowering of the beach face on an adjacent sector |

each of these several factors was considered to have varied spatially and temporally, and Bird advised that 'explanation of erosion should be presented in terms of a ranking of the factors for each coastal sector', considering that 'a single factor explanation usually turns out to be an over-simplification' (Bird 1985, p.158).

Many of these causes of erosion are the result of a negative sediment budget. Although the concept of the coastal sediment compartment and its sand budget had been proposed in the 1970s (Davies 1974; Komar 1976), it became better defined and widely used in the United States in the following decades (Rosati 2005). Coastal sediment compartments, also called littoral or drift cells, were delineated for much of the coast of the United Kingdom and have formed the basis of shoreline management plans for England and Wales (Cooper et al. 2002). A hierarchical system of coastal sediment compartments has more recently been described for the coast of Australia (Thom et al. 2018, Short 2020).

In drawing attention to the prevalence of recession on coastlines, Figure 1 appeared in several publications (Bird 1985, 1987, 1993). This indicated that many sandy coastlines have prograded over recent millennia, forming Holocene beach-ridge plains (also called a relict foredune plain or strandplain), 'which now show evidence of recession on their seaward margins' (Bird 1985, Figure 87, p. 168). Historically, sand has been variously blown onshore (A), moved alongshore (B) or lost to the seafloor (C). Bird (1985) noted that Bruun had argued that a sea-level rise would result in a landward migration of the transverse shore profile (Bruun 1962), and his Figure 88 illustrated what has become known as the Bruun Rule by which a beach, if in equilibrium, would maintain its overall profile,

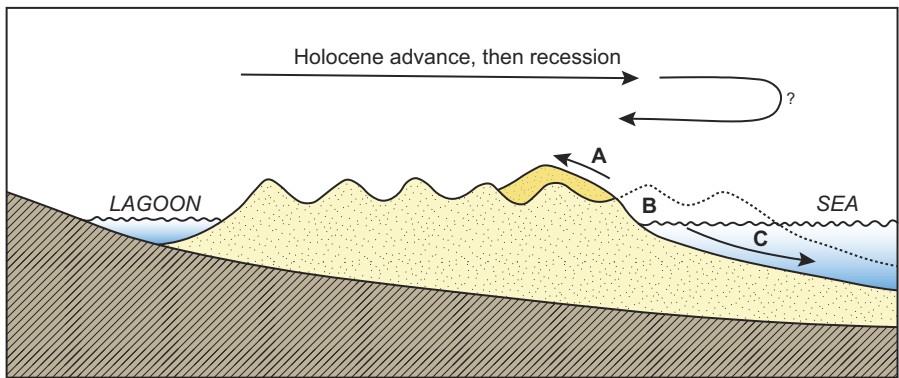

**Figure 1.** A sequence of Holocene prograded beach ridges with evidence of recent recession on the seaward margin. Sand may have been blown onshore (A), moved alongshore (B) or reworked offshore onto the shoreface (C) (after Bird 1985, Figure 87, p. 168).

but be displaced landward in proportion to its nearshore gradient (Bird, 1985, p. 169).

In considering the effects of a rising sea level on coastal environments in his later book, Bird followed the diagram shown in Figure 1 with a fuller explanation of the Bruun Rule, 'in the absence of alternative models' (Bird 1993, p.120). Bruun had proposed a model of the response of a sandy beach to sea-level rise in 1962, anticipating an upward and landward translation of a transverse profile with transfer of sand from the beachface into the nearshore, where an equilibrium existed with no addition or loss of sand (Bruun 1962). It had transformed 'into a "rule of thumb", whereby the coastline retreats 50–100 times the dimensions of the rise in sea level: a 1-m rise would cause the beach to retreat by 50–100 m' (Bird 1993, p. 56).

Preliminary evidence from eroding shorelines on the east coast of the United States and in the Great Lakes, together with laboratory experiments, provided some support for Bruun's hypothesis, and Bruun had further outlined constraints on its use in 1988 (Bruun 1988). Bird also drew attention to several problems with the concept, including difficulties determining an appropriate seaward boundary, or 'closure depth', landward movement of sand to the backshore and the impracticality of ensuring a closed sediment budget, foreshadowing that 'since many seaside resort beaches are no more than 30 m wide, the implication is that these beaches will have disappeared by the time the sea has risen 15–30 cm (i.e. by the year 2030), unless they are artificially replaced' (Bird 1993, p. 56).

A full review of the Bruun Rule is beyond the scope of this study; however, its use remains contentious (Cooper and Pilkey 2004; Cooper et al. 2020).

## Methodological advances since the 1980s

Whereas the heuristic proposed by Bruun before the IGU project remains largely unchanged and widely adopted, primarily because there are still few, if any, alternatives (see below), several methodological advances are considered below, including developments in coastal morphodynamics, geospatial refinements and the increasing potential of remote sensing applications.

### *Coastal morphodynamics and modelling*

Comparing coastlines at two or more instances in time indicates their changeability. However, there is now a much greater

understanding of the co-adjustment of process and form encapsulated in the concept of coastal morphodynamics (Wright and Thom, 1977). Beaches undergo morphodynamic adjustments in response to changes in ambient wave conditions (Wright and Short 1984), comprehensively reviewed by Castelle and Masselink (2023). This more holistic process–response approach to studying coasts triggered the initiation of direct monitoring; for example, surveys of the two Australian beaches described in a later section commenced in the 1970s. Initially, beaches were surveyed using simple techniques, such as the Emery method, using two graduated rods and the horizon (Emery 1961). A suite of different approaches has been adopted to monitor changes at Narrabeen Beach in Sydney (Harley et al. 2011a). Traditional surveying undertaken using a total station or automatic level has been expanded to include GPS profiling, and more complex equipment, such as terrestrial laser scanners, has also been used (Vos et al. 2022), offering monitoring options that can be applied across a range of accessibility and cost (Torres et al. 2024). Video, such as the Argus system, can be used for real-time monitoring to determine shoreline position and wave conditions (Holman and Stanley 2007).

Many coasts undergo quasi-periodic cycles, eroding during winter months but recovering in calmer seasons, or responding to longer-term fluctuations associated with phenomena such as El Niño (Jackson and Short 2020). The use of drones (also called unmanned aerial vehicles) for change detection has increased dramatically in recent years (Casella et al. 2020; Green et al. 2021; Joyce et al. 2023). However, these local-scale high-precision surveys do not extend far enough back in time to adequately assess longer-term trends in accretion or recession.

Morphodynamic adjustments occur over varying spatial and temporal scales. Figure 2 illustrates four scales adopted by Cowell and Thom (1994) to explain past changes on sandy coastlines. The smallest scale covers 'instantaneous' processes of fluid dynamics and sediment entrainment. The 'event' scale involves drivers such as storms, which have a disproportional impact on beach morphology. Evidence of erosion is generally apparent on sedimentary coasts after storms, but many beaches undergo 'cut and recovery', and in the days or months after a storm, sand returns to a beach such that it adjusts towards its pre-storm morphology (Vitousek et al., 2023). It is important to discriminate storm-driven coastal erosion from a longer-term trend whereby the shoreline retreats landwards, a process referred to as recession. Coastal managers, involved with planning, need to consider the 'historical' (or engineering) timescale of several decades over which trends

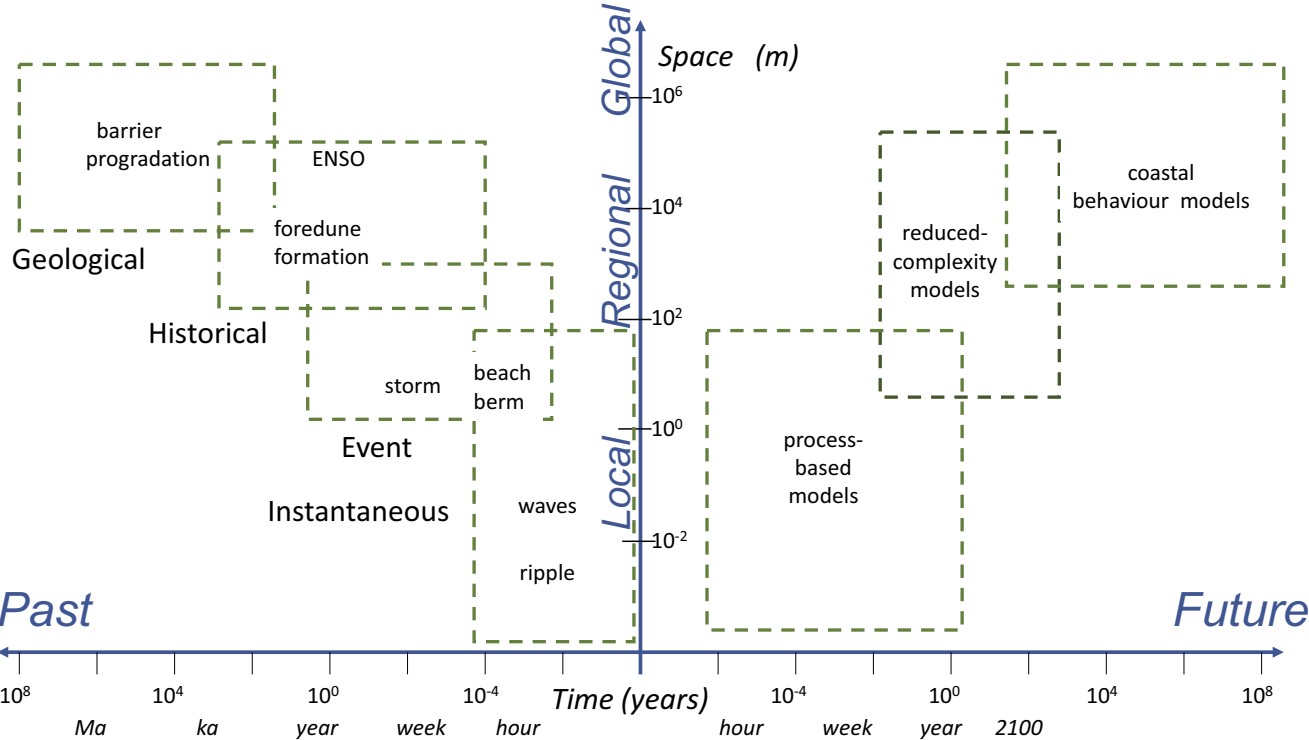

**Figure 2.** A schematic diagram representing spatial and temporal scales relevant to processes on sandy shorelines. The discrimination between instantaneous, event, historical and geological scales follows Cowell and Thom (1994). Representation of future adjustment is shown (following Woodroffe and Murray-Wallace, 2012) with the type of modelling that may be appropriate over these scales.

may become apparent. A longer-term 'geological' timescale can be informed by stratigraphy and dating of sedimentary sequences, such as those contained within the prograded ridges (shown in Figure 1), and may provide insights into net sediment supply.

Figure 2 has been extended to postulate the role of modelling and its potential to provide forecasts of how the coast may behave in the future (Gelfenbaum and Kaminsky 2010; Woodroffe and Murray-Wallace 2012). At a local scale, process models (such as XBeach and Mike 21) may enable simulation of beach adjustment, but they are computationally expensive, and an imperfect representation of physics components leads to aggregation of errors, instabilities and inaccuracies if applied over large areas or beyond days to years. Reduced complexity models (such as IH-LANS and COCOONED) tend to adopt a simpler treatment of wave shoaling and dissipation and sediment transport using conservation of mass/volume, and are designed to be applied at a years-to-decades scale (Hunt et al. 2023). Rule-based large-scale coastal behaviour models (such as the Shoreface Translation Model or GEOMBEST) are designed based on long-term coastal change, considering the overall behaviour of the system (Pang et al. 2023).

### Geospatial advances

The accuracy with which historical comparisons of shoreline position can be undertaken has improved since the 1980s (Burningham and Fernandez-Nunez 2020; Apostolopoulos and Nikolakopoulos 2021), particularly with the adoption of geographical information systems (GIS), which are used to integrate and analyse spatial data or model coastal processes (Sarrau et al. 2024). In an extensive bibliometric review of literature on shoreline change (>1,500

papers), Ankrah et al. (2022) showed how studies have progressed from using simple observations from historical charts and topographical maps to employing high-resolution multi-temporal satellite images. The ability to obtain reliable shoreline change estimates depends on how specific shorelines are represented, whether the horizontal position of a proxy feature is used (such as a waterline or vegetation line) or a datum-based intercept (such as mean sea level) is adopted (Ruggiero et al. 2003).

A major contribution to more rigorous assessment of shoreline change was the development of the Digital Shoreline Analysis System (DSAS, Danforth and Thieler 1992; Thieler and Danforth 1994) and similar approaches (Gomez-Pazo et al. 2022; Mishra et al. 2025), which provide the capacity to calculate shoreline recession rates given a set of mapped shorelines. These tools allow the operator to perform statistical tests, which can be compared with the accuracy of the mapping itself. Where a series of aerial photographic surveys have been undertaken over many years, using DSAS can reveal significant trends in shoreline position (Apostolopoulos and Nikolakopoulos, 2021). Summary measurements, such as the Shoreline Change Envelope, may be useful. If there is a consistent trend, this can be captured using End Point Rate and Net Shoreline Movement, but as these use only the first and last observation from the record, more complex dynamics are better characterised by Linear Regression Rate or Weighted Least Squares Regression.

### Satellite-derived shorelines

In 1985, Bird considered the suitability of satellite imagery for detecting coastline changes but indicated that pixel resolution

was then inadequate for its widespread use. He said, 'undoubtedly techniques of mapping linear features from satellite imagery will improve, but in the meantime conventional air photography has been of much more value in detecting and measuring coastline changes than remote sensing from satellites' (Bird 1985, p. 8). Since then, pixel resolution of satellite imagery has improved substantially, and recent advances in automated algorithms that can extract shoreline positions with sub-pixel accuracy have significantly increased the usefulness of historical satellite imagery for measuring coastal change (Do et al. 2019; Vitousek, et al. 2023). Rapid developments in the acquisition of remote sensing imagery, the making of such imagery accessible under a free and open data policy since 2008 and parallel digital image processing have enabled a range of new global datasets and accessible databases.

Until recently, the Landsat programme has been the principal source for the acquisition of coastal geospatial data for the past three decades, with pixel resolution improving from ~80 to 15 m on the ground. More recently, other satellite missions have been launched, such as the Copernicus Programme operated by European Space Agency, and improvements in resolution will continue over the coming years (Darwash 2024). The various 'Sentinel' missions enable a mid-quality resolution in multispectral bands (10 m) and a frequent revisit time (~5 days). Large spatial-scale data with high temporal frequency are providing considerable opportunities to study coastal morphodynamics (Splinter et al. 2018; Turner et al. 2021; Vitousek, et al. 2023). Hyperspectral sensors are likely to be increasingly used even if, at present, they are limited to exploratory studies (e.g., Souto-Ceccon et al. 2023).

Satellite imagery also allows almost complete global coverage and hence enables worldwide generalisations. This was particularly effectively shown by Luijendijk et al. (2018), who used an automated approach to extract decades of shoreline positions of the world's sandy beaches from global satellite imagery described in the following section. Their quantitative compilation suggested a considerably different pattern to the qualitative assessment undertaken 30 years earlier by Bird.

Analysis of SDSs has progressed significantly in the past few years (Cabezas-Rabadán et al. 2019; Almeida et al. 2021, Pardo-Pascual et al. 2024), with sophisticated methods of processing satellite images facilitating the extraction of high-quality satellite-derived products to detect beach changes (Liu et al. 2017; Pardo-Pascual et al. 2018; Doherty et al. 2022). Two developments in particular have been advantageous: (i) automated shoreline detection algorithms, many at sub-pixel accuracy (Bishop-Taylor et al. 2019; Caldareri et al. 2024), now available as either toolboxes where users extract their own shorelines for their sites (e.g., CASSIE, CoastSat and SAET), or pre-processed datasets (e.g., DEA Coastlines and ShorelineMonitor), and (ii) free archiving of satellite imagery on cloud-based GIS platforms (such as Google Earth Engine) where users can access imagery in an efficient, automated way, as well as do some processing on the cloud without needing to download terabytes of data and process it on their own computer.

The rapid evolution of SDS has been reviewed by Vitousek et al. (2023); >70 studies were published on the topic in 2022. A comparison of five approaches, ShorelineMonitor (Luijendijk et al. 2018), CoastSat (Vos et al. 2019), SHOREX (Sanchez-García et al. 2020), CASSIE (Almeida et al. 2021) and High-tide SDS (Mao et al. 2021) against four sites that have long-term observational beach survey datasets was undertaken by Vos et al. (2023a). At Narrabeen Beach in Australia (see below for more detail), four of the algorithms detected shoreline position to within a horizontal accuracy of 8–10 m. However, accuracy was poorer for a high-energy macrotidal beach, Truc Vert, in France, where only 18% of shoreline change observations fall beyond the 28 m horizontal accuracy, meaning that most of the shoreline variability at this site is drowned in the noise of the satellite time series (Vos et al. 2023a), implying that a customised high-tide algorithm may be more appropriate (Mao et al. 2021; Konstantinou et al. 2023).

## Global trends

In 2018, Luijendijk et al. used an automated approach to extract decades of shoreline positions of the world's sandy beaches at sub-pixel accuracy from global satellite imagery. They said that about 7% of the world's sandy beaches had experienced severe recession. They indicated that their assessment 'shows that 24% of the world's sandy beaches are persistently eroding at a rate exceeding 0.5 m/yr over the study period (1984–2016), while 27% are accreting' (Luijendijk et al. 2018, p. 4). Their study suggested that about 16% of sandy beaches were experiencing erosion at rates exceeding 1 m/year, and 18% were accreting at >1 m/year. They noted that these observations were significantly different from the more qualitative descriptions by Bird (1985, 1987); they also proposed that no single explanation can easily account for observed retreat on any individual beach.

Contrary to the view that 70% of sandy shorelines are experiencing retreat, as expressed by Bird (1985), the analysis by Luijendijk et al. (2018) indicated that there appears to have been accretion on many of the world's coastlines over the past three decades, especially in the northern hemisphere. Coastlines across Eurasia and North America may be changing in more complex ways than those in the southern hemisphere because a greater variety of patterns of relative sea-level change have been experienced, in contrast to the relative stability of sea level over the past 6,000 years at far-field sites, distant from the former Pleistocene ice sheets. Isostatic response to ice loads and their melting since the Last Glacial Maximum can mean that rates of uplift exceed the rate of sea-level rise, constraining patterns of shoreline change (e.g., in much of the Baltic region [Harff et al. 2017]). Consequently, there are fewer locations (Luik et al. 2024 describe one example) that show substantial prograded Holocene coastal plains now prone to erosion (as shown schematically in Figure 1).

A second difference is in the extent to which human actions have modified the coastline (Mentaschi et al., 2018). Coastal intervention works have been implemented on many European shorelines for more than two millennia (Pranzini, 2018). Traditionally, hard engineering measures have been used, such as seawalls, revetments, sea dikes, gabion bags, groynes and breakwaters. These have often been adopted to protect vulnerable infrastructure, mitigating incoming waves and thus reducing erosion, or protecting low-lying areas from inundation. More recently, soft engineering approaches have been increasingly considered, using nature-based solutions that attempt to achieve coastal stability by utilising natural processes and resources (Spalding et al. 2014). Beach nourishment or replenishment, where beaches have been enriched with sand or gravel (Van Koningsveld and Mulder 2004), has also been widely used in the northern hemisphere, and may result in slower long-term rates of retreat, or apparent accretion, countering erosion that might otherwise have occurred due to urbanisation and infrastructure development (Semeoshenkova and Newton 2015; Paprotny et al. 2021).

The EUROSION project quantified coastal erosion in Europe, concluding that ~20,000 km of coastline (notably in Greece,

Cyprus, Portugal, Latvia and Poland) faced serious impacts in 2004, driven by sediment deficits and poorly planned coastal defences, despite protective engineering works on some of them (EUROSION 2004; Monioudi et al. 2017). Rapid coastal development along >100,000 km of coastline in Europe has led to increased coastal risks, exacerbating beach erosion problems particularly in the United Kingdom, Spain and Italy (Cooper et al. 2009). The EUROSION study stressed that the resilience of the coast depends on two key factors: (i) sediments and their redistribution and (ii) accommodation space for retreat of sedimentary systems. It also inferred, as Bird had done, that coastal erosion results from the cumulative impact of a wide range of natural and human-induced factors, none of which may be considered as the single cause of erosion.

Athanasiou et al. re-evaluated the EUROSION study using SDSs and a Bruun-type response to sea-level rise, showing that European shorelines were vulnerable to retreat of 50–100 m by 2100, depending on which sea-level rise projection was adopted (Athanasiou et al. 2020). They recognised that these rates would be modified by variations to the sediment budget and any residual effects of storms and other seasonal, annual or multi-annual fluctuations. Further analysis of satellite-derived long-term coastline changes along European coastlines has suggested substantial differences depending on which optical satellite imagery routines are used, and shows contrasts in some cases with direct observations (Castelle et al. 2024).

Since the 1980s, there has also been a considerable increase in artificially built coastal lands, often euphemistically termed 'land reclamation', particularly in Asia. The coastline of the southern Arabian Gulf, for example, comprised extensive saline mudflats, termed sabkhas, with localised dunes in the 1970s (Bird 1985, p.109). The United Arab Emirates provides a striking example of rapid urban growth with extensive engineering works along several parts of the coast. The population of Dubai expanded from 183,000 in 1975 to over 2 million in 2015, and the land area has increased by >68 km$^2$, despite erosion of up to 30 m/yr on adjacent unprotected shorelines (Subraelu et al. 2022). The appeal of coastal living has seen the city extend since the 1980s, with Palm Jumeira and Palm Jebel Ali constructed in the nearshore and an archipelago of still largely unsettled islands, called 'the World', built offshore (Bonnett 2021).

Urban expansion via land reclamation for 135 cities with populations over 1 million added 253,000 ha to the Earth's surface between 2000 and 2020, primarily for seaport extension (Sengupta et al. 2018; 2020; Sengupta & Lazarus 2023). The coastal zone of mainland China has undergone a significant increase in land area (Wang et al. 2021), with a net increase of about 10,900 km$^2$ from 1990 to 2020 (Li et al. 2023). Recent studies have shown that much newly reclaimed land is facing rapid rates of subsidence of up to 20 cm/yr, and 70% of recent reclamation has occurred in areas identified as potentially exposed to extreme sea-level rise by 2100 (Sengupta et al. 2023).

The regional-global scale of assessment that is now possible using satellite imagery lacks the precision of local-scale studies, and it generally covers only the past three decades, leaving unresolved whether the earlier qualitative summary misinterpreted the proportion of coasts that were undergoing retreat, or whether various anthropogenic interventions have slowed the overall rate of recession, despite an acceleration in the rate of sea-level rise. What is clear is that there are many coastlines where human actions have artificially stabilised shoreline position, and some where land has been formed that was formerly sea. Luijendijk et al. (2018)

estimated net erosion for Australia at an average rate of −0.20 m/yr and also for Africa at a rate of −0.07 m/yr, in contrast to all other continents that showed net accretion. The case of Australia is considered in more detail below.

## Australia: A case study

Australia is located in the far-field, distant from former polar ice sheets and, therefore, with minimal vertical land movement due to glacial isostasy. Bird, who was based in Australia for much of his academic career, used Australian coastal examples in *Coastline Changes* and hypothesised many of these to be dominated by erosion (Bird 1985). Many Australian coastlines comprise a Holocene prograded coastal plain like that shown in Figure 1, with little or no development on it.

Luijendijk et al. (2018) found Australia to be the continent that had experienced the most net shoreline retreat in their assessment of beach erosion based on global satellite imagery. Australia was also identified as especially vulnerable in a forward-modelling study by Vousdoukas et al. (2020), based on an adaptation of the Bruun Rule. Their analysis implied that at least 12,324 km of sandy beach coastline is threatened by erosion, and they considered that about half of Australian beaches would go 'extinct' by 2100.

The entire Australian coast, which has relatively little engineering intervention, has recently been assessed using satellite imagery, particularly Landsat over >30 years, within the Digital Earth Australia (DEA) datacube (Bishop-Taylor et al. 2021). DEA Coastlines combines satellite data with tidal modelling to extract tide-datum-based annual shorelines that represent the typical median location of the mean-sea-level (0 m Australian Height Datum) shoreline for each year from 1988 to present (Bishop-Taylor et al. 2021). The waterline was determined using the Modified Non-Dimensional Water Index for a 30 m-spaced point dataset of derived statistics describing linear regression-based rates of coastline change using these annual shorelines, with a mean absolute error of ~0.35 m/yr.

At the continental scale, 78% of non-rocky coastlines were found to be stable (changing <0.31 m/yr) and 22% were dynamic, with 11% retreating and 11% advancing. Only 0.65% of the coasts were recorded as retreating at more than 5 m/yr (Bishop-Taylor et al. 2021). These observations call into question the erosional trends anticipated by Vousdoukas et al. (2020). Furthermore, an overview of >10,700 beaches has not detected recession at rates implied by their forward modelling (Short 2022). Short has concluded that 'where recession was occurring, it can generally be attributed it to a negative sediment budget, rather than sea-level rise, with major losses to longshore transport and in places inland to dunefields', and that 'there is no evidence to date of rising sea level generating accelerated recession' (Short 2020, p. 178).

Andy Short commenced regular surveys on Narrabeen-Collaroy Beach in Sydney in 1976, and this is now one of the best-understood beaches in the southern hemisphere (Figure 3a). Seasonal surveys, subsequently supplemented by increasingly sophisticated techniques (including Argus coastal imaging, quad-bike Real-time kinematic-GPS, fixed scanning lidar, drone surveys, CoastSnap and satellite-derived techniques; Splinter et al. 2018), indicate that the shoreline undergoes beach rotation primarily related to prevailing wind conditions associated with El Niño-Southern Oscillation (ENSO) (Figure 3b), which influences cross-shore and longshore sand movement, displacing the shoreline by tens of metres (Harley et al. 2011b; Ibaceta et al. 2023). The destructive impact of storm erosion was

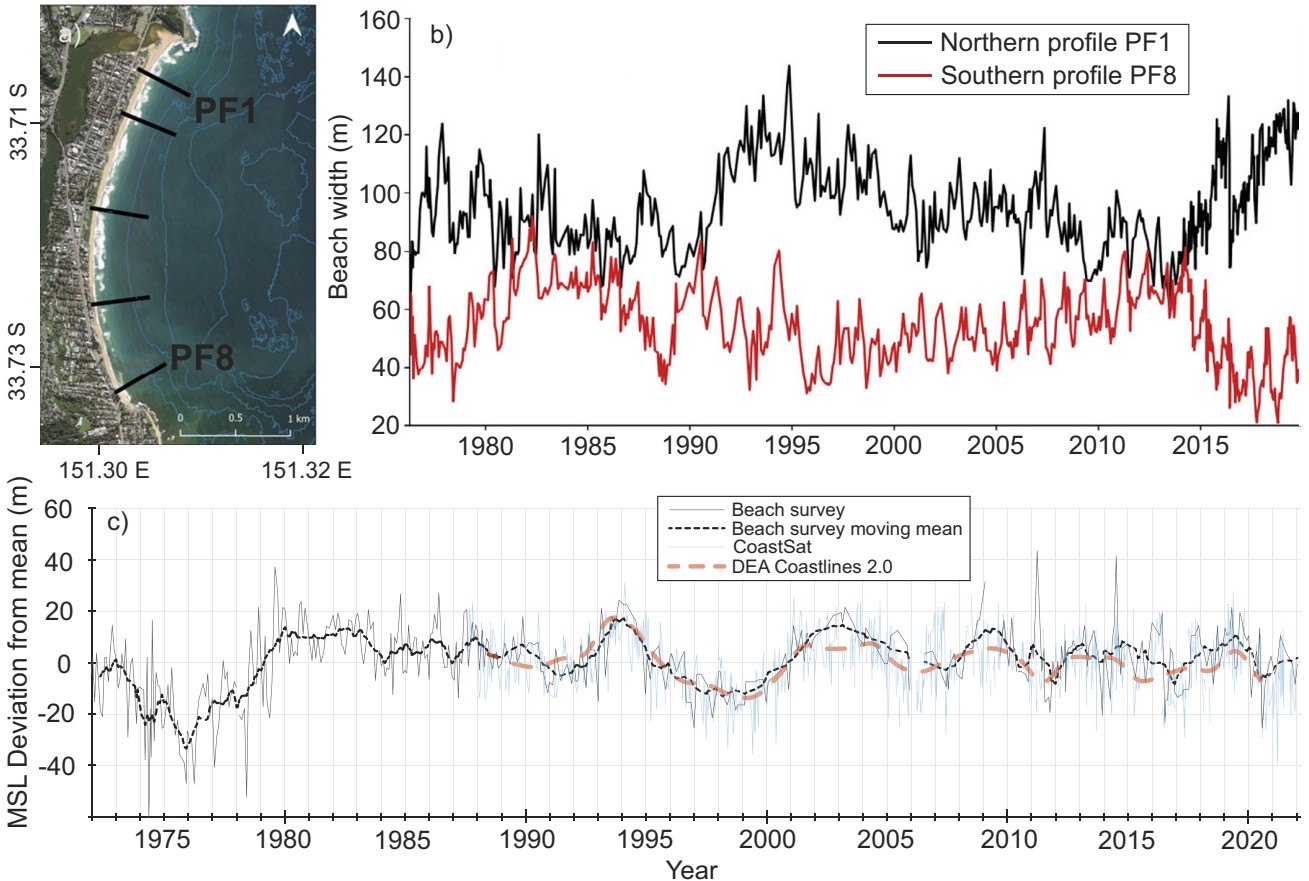

**Figure 3.** Beach behaviour at monitored beaches in New South Wales, Australia. (a) Narrabeen-Collaroy Beach in northern Sydney, and location of monitored profiles. (b) Variation of beach width at northern and southern ends of Narrabeen Beach (see Turner et al. 2016 for the summary of data collection). (c) Variation in beach position for a transect on Bengello Beach, near Moruya, based on a 50-year record of beach and foredune surveys (McLean et al. 2023, see their Figure 7 and Supplementary Material), showing the record from DEA Coastlines and CoastSat for this site for comparison.

demonstrated in 2016 when several beach-front properties were damaged (a backyard swimming pool was undercut and collapsed onto the beach). However, detailed surveys of the beach and near-shore before and after the 2016 event indicate that it recovered to a volume with ~420,000 m$^3$ of sand more than before the storm, an addition of 91 m$^3$/m on average (Harley et al. 2022). The northern end of the beach is as accreted as at any time during the several decades over which it has been surveyed; a response to sea-level rise over the survey period is not detectable.

Narrabeen Beach in the northern suburbs of Sydney is backed by infrastructure, and its behaviour may not be typical of adjacent beaches. By contrast, Bengello Beach, near Moruya in New South Wales, 240 km south of Sydney, is an embayed beach that has been little disturbed. The beach fronts a prograded barrier, similar to that shown in Figure 1, and its Holocene history, examined first using radiocarbon dating (Thom et al. 1981) and subsequently reassessed using optically stimulated luminescence dating (Oliver et al. 2015), indicates that the plain has undergone net progradation at a relatively constant rate over the past 7,000 years.

Beach-foredune surveys at Bengello Beach were initiated by Bruce Thom and Roger McLean and have been maintained several times a year for the past 50 years (Figure 3c). The early surveys captured extensive erosion associated with several storms in May–June 1974, which caused landward retreat of 50–60 m; subsequent recovery to its previous position took several years (Thom and Hall

1991). Since that time, the beach has fluctuated, as shown in Figure 3c (McLean et al. 2023). There has been a net volumetric gain in sand over the past 50 years, derived primarily from offshore, but slightly less than the inferred accretion rate over the late Holocene. McLean et al. caution that 'while it would be premature to infer a slowing of the long-term progradation rate, this comparison is suggestive of such a trend.' (McLean et al. 2023, p. 13). Overwash and destruction of sections of the foredune occurred in 2022 (Oliver et al. 2024), but this prograded barrier does not yet appear to be in the erosional phase implied in Figure 1.

These two long-term records show oscillations of shoreline position but reveal the complexity of coastline changes. The effect of beach rotation (Figure 3b) and successive storms (Figure 3c) masks any incremental response to ongoing sea-level rise at these sites. Although resembling the circumstance represented by Bird in Figure 1, the four to five decades of observations indicate that such beaches undergo a complex sequence of changes, and their behaviour cannot be described by a single simple trajectory.

Observations elsewhere in Australia show the complexity of shoreline adjustment (Nanson et al 2022). Where aerial photograph analyses suggest that formerly oscillating shoreline positions have undergone an abrupt change to recession, these have been attributed to a switch to sediment budget deficit rather than specifically to sea-level rise (e.g., Sharples et al. 2020; Short 2022; Sharples and Watson 2024).

## Discussion

It is frequently stated that a consequence of sea-level rise is likely to be coastal erosion; sometimes this is expressed as accelerated coastal erosion (e.g., Leatherman, Zhang and Douglas 2000; Mimura 2013; Cazenave and Le Cozannet 2013). However, coastal erosion is a natural component of morphodynamic adjustments to changes in ambient wave energy on most beaches (Wright and Short 1984). Seasonal adjustments occur on many beaches, as do cycles of storm cut and recovery (McCarroll et al., 2023). Storms of greater magnitude generally require longer for the beach to regain its former position, if there is no long-term adjustment to the sediment budget. Individual storms can affect adjacent beaches differently, and erosion is likely to vary along the length of the beach.

Recession, indicated in the statement that 70% of sandy shorelines have been retreating, may result from a rise in sea level, but it may more often imply that there is a negative sediment budget at a particular site. The estimate that 70% of the world's sandy shorelines had been retreating before 1985, proposed by Bird (1987), no longer appears tenable. Despite the extensive list of contacts and the numerous case studies compiled in that assessment, it was primarily a qualitative estimation. Methodological advances in the four decades since the compilation by the *Commission on the Coastal Environment* have enabled more detailed quantitative assessments of shoreline trajectories at many sites.

It has become apparent that oscillations of the shoreline are often part of a broader pattern of recurrent changes (Camfield and Morang 1996) – for example, Narrabeen Beach in Australia shows rotation that is driven primarily by ENSO. ENSO has been shown to be an important constraint on beaches across the Pacific Ocean (Vos et al. 2023b) and may also affect beaches at a more global scale (Almar et al. 2023). In numerous instances, the natural variability of shoreline changes seems likely to overwhelm and, hence, mask what adjustment might be attributable to sea-level rise alone (Banno 2023).

The compilation by Luijendijk et al. (2018) from recent decades of satellite imagery has revealed that beaches on many continents have remained stable or accreted, and only a quarter of the world's beaches have undergone detectable recession to date (at >0.5 m/yr). The use of satellite imagery at national and regional scales provides insightful comparisons of the relative magnitude of change, but these broader scales may not be representative of local complexities or anomalies visible at finer resolutions (Vitousek et al. 2023). Undoubtedly, retreat would have been more widespread on coastlines were it not for extensive engineering intervention works along many coasts, and numerous locations where the coast has been extended by reclamation.

The DEA compilation of ~30 years of primarily Landsat-derived imagery indicates that only 11% of Australia's sandy coastlines have been eroding (at the level of detection, ~0.3 m/yr), but these mean rates of change mask considerable temporal variability (Bishop-Taylor et al. 2021). Sub-annual adjustments are not captured by the annually averaged approach, and complexities of overlapping shorelines associated with mobile shoals, sediment bars, recurved spits and other landforms sub-parallel to shore still limit the extent to which change can be discriminated with sufficient spatial resolution in some geomorphological settings (Nanson et al. 2022). Figure 3c shows the DEA annual shorelines for Bengello Beach plotted in comparison to regular beach surveys; the general pattern of variation is captured, but details of sub-annual variations are not detectable in the DEA annual shorelines. Also plotted in Figure 3c is the CoastSat record for this site, which does capture oscillations like those in survey data, although as the dates of in-field survey and those of satellite overpasses do not coincide, there remains considerable variability. Satellite data were not available for the major erosional storm events that occurred along the New South Wales coast in 1974, so the substantial erosion and the gradual recovery from such extreme events are not incorporated in modelling based only on the past three decades.

The SDS analysis by Luijendijk et al. (2018) also indicated net erosion for Africa. The recent compilation of Digital Earth Africa with a similar Coastlines product to that for Australia offers the opportunity to look in more detail at patterns of shoreline change over the >30 years of Landsat and later satellite imagery. We are not aware of a quantitative compilation of relative accretion and retreat rates from that dataset.

### Oscillations and trends in beach behaviour

It remains challenging to accurately measure, monitor or model consistent shoreline proxies over appropriate spatial and temporal scales, particularly for highly dynamic coastlines. Identification of a trend on any beach is highly dependent on (i) the time over which it is considered, (ii) the technology (image resolution) and proxy (vegetation line and water line) that are used and (iii) other factors (location and geomorphology). The choice of a proxy for the shoreline, reviewed by Boak and Turner (2005), remains challenging; trends shown by proxies, such as mean-sea-level intersect, toe of dune and vegetation line, often vary from each other (Ruggiero et al. 2003).

Beaches are dynamic, and comparisons of shoreline position need to be undertaken over long periods of time to determine trends and to assess whether there is retreat, as opposed to simply local erosion, which is likely to recover with no net loss of sand. Coastal vegetation can play a fundamental role in mitigating coastal hazards by attenuating wave energy (Vuik et al., 2016; Wang et al., 2017). Where vegetation has been planted, such as for dune remediation, it can stabilise sand, reducing coastal erosion (Walles, 2015), and protect landward areas from the effect of storm waves, as well as coastal flooding (Evelpidou et al. 2022). Vegetated foredunes can be reworked several times by storms and migrate with rising sea levels (Ollerhead and Davidson-Arnott 2022). In the absence of severe human impacts, coastal dunes are resilient eco-geomorphological features that can adapt to changes in sea level (Davidson-Arnott and Bauer 2021) and provide a range of ecosystem services and natural habitats (Walker et al. 2017). It is too simplistic to infer an erosional trend from a dune scarp; recovery is likely as sand is returned to the beach and dune following individual storms (Phillips et al. 2019).

### Coastal recession and sea-level rise

There is a wide perception that a rise in sea level will lead to the displacement of the shoreline landwards (Pang et al. 2023). The waterline can be seen to migrate landwards as the tide rises on a sloping shore. This has been formalised into what is called the Bruun Rule, proposed by Per Bruun in 1962 (Bruun 1962). The Bruun Rule predicts that the net transport of sand with rising sea levels is offshore. It has been used to predict coastal erosion with rising sea levels, but its indiscriminate application has been contested (Cooper and Pilkey 2004; Davidson-Arnott 2005).

It is overly simplistic to anticipate that coastal recession could be directly attributed to sea-level rise without considering the various

other factors that affect a beach, particularly those identified by Bird (1987), but also an increasing trend in wave height and changes in storminess (Young and Ribal 2019; Bernier et al. 2024). Many of the factors relate to the source, transport and sinks of sand, emphasising the importance of understanding the sediment budget to explain the pattern of sediment losses from a beach system (Thom et al. 2018). For these, and many other reasons, it remains necessary to consider sections of coast individually before attributing changes to alteration in relative sea level, climate, geomorphology or human actions.

Evidence suggests that natural coastlines can adapt to subtle changes in sea level; they continually adapt, on a regular basis, to much larger disturbances, as demonstrated in a review of adaptations of coastal environments to water-level changes on both ocean and lake shorelines (Davidson-Arnott and Bauer 2021). These authors argue that the net transport of sand due to rising sea levels is onshore, in line with other work on the nearshore (Aagaard and Sorenson 2012) and coastal dunes (Ollerhead et al. 2013). Loss of sand seawards may occur on steep beach-nearshore profiles, but coastal recession can be more rapid on low gradient coasts, such as those with barrier islands (Cowell et al. 2006). Such low-elevation barriers experience landward movement of sand by overwash and barrier rollover (Thomas et al. 2024); they are more likely to retreat in response to sea-level rise, although perhaps with a lag (Cowell and Kinsela 2019; Mariotti and Hein 2022). Simple heuristics, such as the Bruun rule, over-simplify shoreline morphodynamics; other processes shaping the coastline also need to be considered (Cooper et al. 2020). Simulation models are being developed, building on the shoreface translation model (Cowell et al. 1992), which allows for a wider range of sediment transport responses on different types of coasts (McCarroll et al. 2021, 2025). The various numerical modelling approaches to predicting shoreline and coastal morphological change over decadal timescales are reviewed by Hunt et al. (2023).

The concept behind the Bruun Rule applies to an averaged, or equilibrium, beach and nearshore profile, which might be anticipated to adjust such that there is an upward and landward translation of this averaged profile in response to a generally higher sea level (Bruun, 1988). However, beaches are actively adjusting around this hypothetical equilibrium morphology; the Bruun Rule appears unsuitable for local-scale assessments in which reliable estimates of recession are required (Ranasinghe 2016). The considerable variability of monitored beaches, such as those shown in Figure 3, often overwhelms any subtle adjustment to a scarcely perceptible several-centimetre rise in sea level over recent decades (Banno 2023). The most important reason for long-term erosion is often a deficit in the sediment budget, necessitating a consideration of the movement of material and what losses or gains there are for any stretch of coastline over a range of timescales. Coastal squeeze by human infrastructure is also a major issue in the face of rising sea levels (Davidson-Arnott and Bauer 2021).

## Prospect

Our understanding of coastal processes is still based on selective studies of limited parts of the world's coastlines, as it was when the summary was published by the Commission on the Coastal Environment (Bird 1985). Recent advances in the interpretation of shoreline change from satellite imagery and other technologies offer the potential for wider regional, and even global, assessments of broad trends and will become an increasingly important component of interpretation. When coastal recession is observed, it will be important to consider the range of potential explanatory factors, such as those listed in Table 1.

The studies of retreating coastlines undertaken in the 1970s and 1980s and summarised by the Commission emphasised a range of factors that contributed to erosion of beaches; sea-level rise was one of them, but the rate at which sea level was rising was not an issue of particular concern at the time. During the four decades since these studies, there has been wider recognition of the increasing rate of sea-level rise, but it remains difficult to determine the extent to which recession of any shoreline can be attributed to sea-level rise. Successive assessments by the Intergovernmental Panel on Climate Change have stressed that sea level is committed to rise for centuries due to ongoing ocean warming and ice melt. Seasonal and inter-annual variability is likely to be the principal driver of coastline change in the coming decades, although sea-level rise may exert a more detectable effect in the second half of the twenty-first century (D'Anna et al., 2022; Hunt et al., 2023). Sea-level rise will increasingly compound the factors already contributing to coastal behaviour; it is likely to exacerbate erosion, and it will result in more widespread coastal flooding.

Advances in technology, particularly broad-scale SDSs and local-scale survey and terrain modelling, are rapidly improving the capacity to monitor shoreline change. Modelling offers the potential to extrapolate morphodynamic trends into the future. Physics-based models are already applied at the local scale for short timescales. Reduced-complexity models offer potential to foreshadow changes over decadal timescales, although a recent comparison of the performance of five such models applied to the data-rich Narrabeen Beach (Figure 3) showed that their accuracy varies significantly depending on the area evaluated and local conditions (Repina et al. 2025) reflecting the complexity inherent in the prediction of coastal evolution.

Location-specific knowledge of coastal dynamics will continue to be required to enable more sustainable coastal human–environment interactions in the face of climatic and societal challenges. Similar factors are likely to apply to the erosion of cliffs, to changes in muddy coastlines and coastal wetlands, and to the vulnerability of small islands such as those on coral reefs, which have been outside the scope of this overview. However, the principal challenge may not be so much for natural environments, which are likely to have the capacity to change and adapt, but for coastal systems where adjustment is constrained by anthropogenic disturbance and infrastructure.

**Open peer review.** To view the open peer review materials for this article, please visit http://doi.org/10.1017/cft.2025.10010.

**Data availability statement.** No data were created during this overview.

**Acknowledgements.** This study was initially conceived during the centenary of the International Geographical Union in 2022. It is an outcome of a reconsideration of the conclusions of the IGU Commission on the Coastal Environment (1972–1984) by several members of the subsequent IGU Commission, the Commission on Coastal Systems. The present manuscript is based on an oral paper delivered at the 35th International Geographical Congress in 2024 in Dublin. We thank Tom Oliver, Oxana Repina and two anonymous reviewers for constructive comments.

**Author contribution.** All authors have made contributions to this submission. CDW: Conceptualisation, writing, review and oral presentation. NE: Conceptualisation and writing. IDF: Writing and review. DRG: Writing and review. DS: Writing and review. AK: Writing and review. PC: Review and editing.

**Financial support.** No direct funding was received in preparation of this overview, but support from the International Geographical Union to the Commission on Coastal Systems enabled discussions, and DS acknowledges funding from the European Space Agency (ESA) under the WIDGEON-Waterborne Infectious Diseases and Global Earth Observation in the Nearshore.

**Competing interests.** The authors declare none.

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
