## [Reviewer Report]

Coastline changes: a reconsideration

Woodroffe et al.

General

I really like the premise behind this paper – a re-evaluation of the very influential reviews of the IGU’s Commission on the Coastal Environment in the 1970s and early 1980s and where we are now - and it seems very appropriate for this new journal. I really want to strongly support the approach and the paper. In an intellectual environment where authors often claim to be the first to do something, it is really nice to see the historical contexts and more could be made of this. But I think the paper needs quite a bit of work to make it publishable. The depth of analysis (even the dates of the literature cited) vary markedly from section to section and there is repetition of material across some sections. Some material is standard textbook material which lacks focus on the theme of coastal change. Were different sections supplied by different authors and then clipped together? Perhaps that is very unfair but it has that ‘written by Committee’ feel to it at present. The paper seems to struggle as to what it is trying to do, flipping between historical assessment, general coast processes text and descriptions of methods, often in close juxtaposition to one another. But it does get to a sensible set of conclusions. A re-organised and tightened paper would make a very useful contribution to the literature so I encourage revision of this submission most wholeheartedly.

In the first half of the paper why not do a full assessment of the work of Bird and a state of coastal science that it represents? And set that more firmly in a ‘Bird believed that’ set of arguments (perhaps with some direct quotes?) and writing style. The paper should include the discussion of how the Bird data was obtained from the network of correspondents here and some idea of geographical coverage to set alongside the post-1985 methodological advances. And there are some ‘elephants in the room’ here which don’t get full assessment. I agree that the sea level rise debate was still young but it seems odd not to discuss the Hoffman EPA Reports of the 1980s. Why did Bird only pick up on this in the 1990s (when very different sea level scenarios were starting to emerge in the IPCC Reports)? Were the 1980s outputs of the Commission themselves historical records of data gathering in the 1970s (i.e. the state of coastal geomorphology pre-SLR? – interesting in itself). Surely you have to say more about how the ‘Bruun Rule’ from the 1960s was re-invigorated in the 1970s and 1980s in the context of sea level rise, seeming to offer a very simple way to translate rates of sea level rise into rates of shoreline retreat? (and the Bruun Rule is a problem that will not go away – see below). It is interesting to read that Bird (1987) had a very balanced view of the reasons for shoreline retreat than simply being due to sea level rise which is a viewpoint that is coming out in more recent SDS research and in criticisms of simplistic GEE assessments of global shoreline change which point to the important of local controls. Have we come full circle over a period of 40 years, to a renewed realisation that sandy beach erosion it is not solely about sea level rise? There is also the question of ‘coastal morphodynamics’ and its evolution in the 1970s and 80s. That is hinted at in Figure 1. You reference Wright and Short (1984) but the classic paper was by Don Wright and Bruce Thom in the first volume of Progress in Physical Geography in 1977 and this is not referenced. That really did come to define a completely new approach to coastal dynamics, with its coupled space-time hierarchy of coastal change.

The sections on post-1985 research are very mixed, at worst very generalised and uncritical but at best – as in the justifiably long set of Australian case studies – showing real understanding and criticality.

Detail

Pages 4 and 5

The Introduction can start at ‘In 1972…’ ; the first 6 lines add little, particularly to readers of this journal. The third paragraph could be omitted here – there isn’t much detail and I don’t think it fits with the Introduction. The bottom of Page and the top of Page 5 seems very odd to me – there is clear repetition from earlier on page 4 and I don’t see why there should be a focus on the UK, including general text on UK coastal erosion rates; this material could be largely omitted and the odd point integrated into the text that comes before. Under ‘Factors contributing to coastline changes’ tighter focus on the Bird Table would be helpful; too much of the paragraph beginning ‘In a summary…’ reads as just general coastal textbook.

Page 7

Is this text in the correct place? Bird himself placed these questions very early in the 1985 book. I think this is right. Discuss methods first and then results. There is a considerable literature on shoreline proxies and it would be helpful to flag up proxy-based v. datum-based shorelines at some point. I think it is worth making the point here that the intertidal and shallow subtidal zones are highly problematic for mapping purposes – land surveyors don’t go into the water and vessels avoid very shallow water.

Page 8, Page 9 and Page 10

There is another ordering issue here. You need satellite imagery first (data source) before DSAS (data analysis). Why not provide a table of the different satellite platforms, when they came on stream and associated improvements in resolution? (one key date, for example, is March 1978 and Landsat-3). What is the sub-pixel accuracy that you refer to? There is better detail here - methods (but where is the ‘Argus’ system of videography (e.g. Holman and Stanley (2007))) and publication effort (but add Konstantinou et al. (2023) and Castelle et al. (2024)) - but some critical analysis is badly needed, particularly of the landmark paper by Luijendijk et al. (2018), and subsequent related papers (e.g. Vousdoukas et al. (2020)). There are serious criticisms of these approaches, including the continued use of the Bruun Rule (e.g. Cooper et al. (2020)). There are more serious questions here than the statements at the start of ‘emerging trends’.

Page 10, Page 11, Page 12, Page 13, Page 14

‘Coastal engineering interventions’ and ‘land reclamation’ lose all focus on the coastal change issue and read like material from a general coastal textbook. I cannot see the value of this material in the context of this paper.

But the material on Australia is much better in terms of detail and depth of argument and here we do see good criticism of Luijendijk et al. (2018) and Vousdoukas et al. (2020).

Page 15 and Page 16

There isn’t the same level of detail or argument for either Africa, South America or Europe. Again this reads like general text. There are just a few estimates of rates and patterns of coastline change.

Page 17 and Page 18

Perhaps over long and could be tighter but some good points here.

Page 19

The discussion of the Bruun Rule could (should?) be more hard hitting and could engage with the wider literature on it here.

Page 20

One thing that you don’t really mention is the potential for increased storminess in the near-future which has become a greater element of coastal change studies in the last 5 years. It at least deserves a mention.

---

## [Reviewer Report]

This paper addresses a timely and important topic: the long-term evolution of sandy coastlines in the context of sea-level rise. The authors revisit the widely cited assertion from the 1980s, based on the International Geographical Union’s Commission on the Coastal Environment, that over 70% of the world’s sandy coastlines are eroding. Drawing on advances in remote sensing, particularly satellite-derived shoreline (SDS) data, the paper challenges this generalization and highlights the increasing evidence that many sandy beaches are stable or even accreting. I strongly agree with the two key messages the authors aim to convey: (1) the commonly held belief that most sandy coasts are experiencing widespread, long-term erosion is increasingly questionable, and (2) the current impacts of sea-level rise on global shoreline trends are not yet clearly discernible, given the small magnitude of rise and the complexity of shoreline dynamics.

While these key points are valuable and timely, I find the overall structure and focus of the paper to be lacking, which unfortunately weakens the delivery of its core message. The manuscript suffers from frequent shifts between spatial and temporal scales, which create confusion rather than building a coherent narrative. For example, the authors frequently alternate between global, regional, and site-specific discussions, which disrupts the logical flow. A tighter structure, focused more directly on shoreline trends as observed through ~40-yr satellite-derived shoreline (SDS) and other long-term datasets, for instance historical aerial photographs or the handful of intensively monitored study sites, would improve the clarity and impact of the paper.

One major issue is the inclusion of content that, while technically accurate, appears somewhat out of scope. For instance, discussions around UAV and airborne lidar technologies, though relevant for high-resolution, local monitoring, are not particularly useful in a discussion centered on long-term, large-scale shoreline change. Their inclusion, and some others, detracts from the main argument and contributes to the sense of a paper that is overextended. Similarly, the regional breakdown into sections for Africa, Australia, Europe, etc., seems unnecessary unless there is a specific, quantitative reason to highlight each region individually. These sections do not provide new insights that justify their length and may be better consolidated into a brief global synthesis.

Another notable weakness lies in the discussion of uncertainties associated with satellite-derived shorelines. While the authors acknowledge the potential errors in SDS datasets, they largely overlook the distinction between absolute positional uncertainty and trend uncertainty. This is a an important omission. Recent studies have shown that while individual shoreline positions may carry some noise or bias, the use of long, dense time series allows for good trend extraction through linear regression or similar methods. The real question is not whether the data are noisy, but whether the trend, often based on hundreds or even thousands of observations, is statistically sound and physically meaningful given that potential long-term trend bias have been identified (see Vos et al., 2024 and others). This is missing from the paper.

The authors also underrepresent a body of literature on the delayed emergence of the sea-level rise signal in shoreline trends. It is now well established that natural shoreline variability at seasonal to interannual timescales often masks the more gradual SLR signal. Modelling studies have shown that the impact of SLR on sandy shorelines, particularly in the context of climate change, is unlikely to emerge clearly from background variability until the second half of this century. This literature is highly relevant to the central question addressed in the manuscript but is disregarded. Including this perspective would greatly strengthen the argument that current trends are dominated by sediment budget dynamics, storm-driven changes, and other local factors rather than by SLR.

Lastly, I find that the title and overall framing of the paper would benefit from more precise alignment with the actual content. While the manuscript’s focus is almost entirely on sandy beaches, it briefly mentions other coastal environments. If the paper is intended to be about sandy coasts, as the analysis clearly is, then this should be reflected more clearly in the title and scope, rather than suggesting a broader coastal generalization.

In summary, this paper offers an important re-evaluation of common assumptions about global coastal erosion. However, the manuscript in its current form is too diffuse and occasionally off-topic, which diminishes its potential impact. I recommend a major revision to refocus the narrative around the most relevant datasets and concepts, streamline the structure, and engage more with recent literature on trend uncertainty and detectability of SLR impacts. I acknowledge that editorial decisions ultimately rest with the journal, and I would not object if the paper were eventually accepted in its current structure. However, I believe a more concise and focused version would better serve the scientific community and the important message the authors seek to convey. I hope my comments can be of help to the authors.

---

## [Editor Report]

Dear Prof Woodroffe and co-authors,

Thank you very much for your submission and many apologies for the very long wait for the review process to complete. 

As you will see from the detailed reviews submitted, both reviewers clearly value this contribution to this Special Issue and have taken particular care in reviewing the content with some very constructive suggestions. They particularly value the key messages around the complexities of the coastal system response to sea level rise and the contradictory simplification of sea level rise response in much of the international literature at different stages over the history of sea level research. The reviewers also agree around some of the short-comings in that both comment on the rather patchy quality and at times confusing structure and emphasis within the paper. 

Given the strong agreement between the two reviewers around the relevance of this paper to the special issue, I very much hope that their constructive suggestions which, to my mind, very much complement each other, can be taken into account by the team of authors and a revised version can be submitted. The paper is very timely, relevant, and will no doubt stimulate discussion within the international community - however, the comments made by both reviewers justify major revisions to allow the paper to have the impact it deserves to have and I encourage the authors to submit after carefully addressing both reviewers' points on structure and content.

---

## [Reviewer Report]

The revision of this manuscript has been extremely thorough. The authors have taken on board just about every query raised by both Reviewer 1 and Reviewer 2, both those relating to content and to those related to the structuring of the material (including the removal of some less relevant material). Where the authors don’t make changes asked for they argue convincingly why not. The introduction of material on coastal morphodynamics is important and the new Figure 2 valuable. discussion of the Bruun Rule is now better organized. I think that the discussion of the original Luijendijk et al. (2018) could be more critical and there are still a few lines that don’t fit so well. But overall, looking at the revision alongside the original submission, this is now a much better structured and tightened piece that reads very well. It should now be accepted for publication.

---

## [Reviewer Report]

Thank you for considering my suggestions during the first round of review. I appreciate the authors' efforts to tighten and focus the paper. However, I believe a more substantial trimming would still be beneficial. For instance, there remains an entire paragraph on drones, which seems out of place in a review focused on long-term shoreline trends and SLR. There are still a few such examples like this throughout the paper. While they do not fundamentally detract from the work, they do make it harder for the reader to extract the key material efficiently. Having the authors making a last effort on this would be welcome. This is not a necessary requirement, but just my own feeling that the paper would receive more attention this way.

I have one final comment, which should be straightforward to address and would not require another round of review. As the paper aims to cover a wide range of topics, some subsections feel unbalanced, discussing certain aspects in detail while overlooking others. In this context, referencing recent comprehensive reviews would help strengthen the manuscript and better guide the reader. Several such reviews have been published in Cambridge Prisms: Coastal Futures, for example:

Subsection « Coastal morphodynamics and modelling »: Castelle and Masselink (2023, Cambridge Prisms) offer a solid overview of beach morphodynamics and associated temporal cycles. Hunt et al. (2023, Cambridge Prisms) provide a very good overview of the different modelling approaches. I found this subsection « Coastal morphodynamics and modelling », particularly on the modelling side, and still containing irrelevant material, such as drones. Anchoring this section in the aforementioned reviews would help frame the discussion around key concepts and improve its clarity.

Subsection « Satellite-derived shoreline »: This subsection would benefit greatly from drawing on Vitousek et al. (2023, Cambridge Prisms), which provides a comprehensive review of the topic.

These key references should be cited either at the beginning of the relevant subsections to provide context, or at the end, to guide readers seeking a more comprehensive overview. Currently, some of these important review papers are only briefly mentioned mid-paragraph, alongside more specific studies, which makes them less visible to the reader.

Finally, on page 19: The third paragraph on shoreface translation is nicely done. I suggest adding a reference to the ShoreTrans model (McCarroll et al., 2021, Marine Geology), one of the first modelling efforts to go beyond the Bruun Rule and capture additional shoreline translation modes.

---

## [Editor Report]

Both reviewers agree that the authors made substantial changes in light of the earlier reviewer comments. These changes almost fully addressed all points raised by the reviewers. There remain just some very minor revisions that are suggested particularly by reviewer 2 and I feel these will significantly strengthen the paper and allow readers to appreciate the wealth of geomorphological knowledge that underpins the points made by the authors. My recommendation would thus be for the authors to just go through one very small further edit, incorporating the references suggested by reviewer two into the manuscript. I agree with reviewer 2 that these changes are very straightforward and I very much hope that the authors would agree and be able to easily incorporate those. They are listed again here below:

Subsection « Coastal morphodynamics and modelling »: Castelle and Masselink (2023, Cambridge Prisms) offer a solid overview of beach morphodynamics and associated temporal cycles. Hunt et al. (2023, Cambridge Prisms) provide a very good overview of the different modelling approaches. I found this subsection « Coastal morphodynamics and modelling », particularly on the modelling side, and still containing irrelevant material, such as drones. Anchoring this section in the aforementioned reviews would help frame the discussion around key concepts and improve its clarity.

Subsection « Satellite-derived shoreline »: This subsection would benefit greatly from drawing on Vitousek et al. (2023, Cambridge Prisms), which provides a comprehensive review of the topic.

These key references should be cited either at the beginning of the relevant subsections to provide context, or at the end, to guide readers seeking a more comprehensive overview. Currently, some of these important review papers are only briefly mentioned mid-paragraph, alongside more specific studies, which makes them less visible to the reader.

Finally, on page 19: The third paragraph on shoreface translation is nicely done. I suggest adding a reference to the ShoreTrans model (McCarroll et al., 2021, Marine Geology), one of the first modelling efforts to go beyond the Bruun Rule and capture additional shoreline translation modes.

---

## [Editor Report]

I am very grateful to the authors for taking the reviewer comments on board and am very happy to recommend acceptance of the manuscript. I would particularly like to thank the authors also for their patience during the review process and for choosing this special issue for their publication.